# Cornflower Extract and Its Active Components Alleviate Dexamethasone-Induced Muscle Wasting by Targeting Cannabinoid Receptors and Modulating Gut Microbiota

**DOI:** 10.3390/nu16081130

**Published:** 2024-04-11

**Authors:** Ngoc Bao Nguyen, Tam Thi Le, Suk Woo Kang, Kwang Hyun Cha, Sowoon Choi, Hye-Young Youn, Sang Hoon Jung, Myungsuk Kim

**Affiliations:** 1Natural Product Research Center, Korea Institute of Science and Technology, Gangneung 25451, Republic of Korea; 220530@kist.re.kr (N.B.N.); 025365@kist.re.kr (T.T.L.); kangsw@kist.re.kr (S.W.K.); soun10@yonsei.ac.kr (S.C.); hyyoun@kist.re.kr (H.-Y.Y.); 2Department of Biochemistry and Molecular Biology, College of Dentistry, Gangneung Wonju National University, Gangneung 25451, Republic of Korea; 3Natural Product Informatics Research Center, Korea Institute of Science and Technology, Gangneung 25451, Republic of Korea; chakh79@kist.re.kr; 4Division of Bio-Medical Science and Technology, KIST School, University of Science and Technology (UST), Daejeon 34113, Republic of Korea; 5Department of Convergence Medicine, Wonju College of Medicine, Yonsei University, Wonju 26426, Republic of Korea

**Keywords:** *Centaurea cyanus*, muscle atrophy, dexamethasone, cannabinoid receptors, gut microbiota

## Abstract

Sarcopenia, a decline in muscle mass and strength, can be triggered by aging or medications like glucocorticoids. This study investigated cornflower (*Centaurea cyanus*) water extract (CC) as a potential protective agent against DEX-induced muscle wasting in vitro and in vivo. CC and its isolated compounds mitigated oxidative stress, promoted myofiber growth, and boosted ATP production in C2C12 myotubes. Mechanistically, CC reduced protein degradation markers, increased mitochondrial content, and activated protein synthesis signaling. Docking analysis suggested cannabinoid receptors (CB) 1 and 2 as potential targets of CC compounds. Specifically, graveobioside A from CC inhibited CB1 and upregulated CB2, subsequently stimulating protein synthesis and suppressing degradation. In vivo, CC treatment attenuated DEX-induced muscle wasting, as evidenced by enhanced grip strength, exercise performance, and modulation of muscle gene expression related to differentiation, protein turnover, and exercise performance. Moreover, CC enriched gut microbial diversity, and the abundance of *Clostridium sensu stricto 1* positively correlated with muscle mass. These findings suggest a multifaceted mode of action for CC: (1) direct modulation of the muscle cannabinoid receptor system favoring anabolic processes and (2) indirect modulation of muscle health through the gut microbiome. Overall, CC presents a promising therapeutic strategy for preventing and treating muscle atrophy.

## 1. Introduction

Skeletal muscle, accounting for a substantial 30–40% of body mass, plays a crucial role in movement and metabolism [1]. Yet, its integrity can be jeopardized by protein imbalance, leading to muscle-wasting syndromes like sarcopenia and cachexia. These debilitating conditions have profound impacts on patients suffering from various disorders, including kidney failure, diabetes, and advanced cancer, severely compromising their physical function and quality of life [2,3]. Factors contributing to muscle atrophy are diverse, encompassing physical inactivity, aging, injuries, and a range of diseases like obesity, diabetes, and cancer [3]. Among these, glucocorticoids, widely used for treating inflammatory and autoimmune conditions, are notorious for inducing muscle loss. Dexamethasone (DEX), a potent synthetic glucocorticoid, exemplifies this adverse effect, with high doses causing sarcopenia in up to 60% of long-term users [4].

High-dose DEX administration disrupts muscle homeostasis primarily by hindering protein synthesis, activating protein degradation pathways, and interfering with muscle cell signaling [5]. This leads to increased expression of muscle-wasting genes like MuRF1 and Atrogin-1 and decreased activity of crucial protein synthesis regulators like mTOR, p70-S6K1, and 4E-BP1 [6,7]. Notably, DEX-induced muscle wasting is further exacerbated by mitochondrial dysfunction, with dysregulation of peroxisome proliferator-activated receptor gamma coactivator 1-alpha (PGC-1α), a central regulator of mitochondrial biogenesis, playing a crucial role [8,9]. In addition to the above-reported mechanisms, therapeutics for muscle atrophy using medical food mixtures or targeting growth hormone, nuclear receptors, angiotensin II receptors, or androgen receptors are in clinical trials, but none have yet been approved by the FDA [10]. This underscores the critical need to explore novel therapeutic targets and strategies to effectively and safely prevent muscle atrophy.

Emerging research shines a light on the gut–skeletal muscle axis, where gut microbiota composition significantly impacts muscle function and mass. This intriguing connection offers a promising avenue for preventing and treating muscle atrophy [11,12,13]. Studies have shown that specific gut bacteria like *Lactobacillus* spp. and *Bifidobacterium* spp. significantly improved muscle mass, strength, and endurance capacity in aged mice [14,15]. In addition, the intake of SCFAs, gut microbial metabolites, prevents muscle strength decline in humans and mice [16]. Therefore, promoting the growth of specific microbes associated with muscle function or increasing the production of microbial metabolites may be an effective treatment strategy to mitigate muscle wasting.

In a preliminary study, we screened approximately 300 natural product extracts whose effects on muscle atrophy have not been previously reported in the literature for their ability to enhance myofiber width in DEX-induced C2C12 myotubes and finally selected *Centaurea cyanus* L. as a candidate in terms of efficacy and extraction yield. *Centaurea cyanus* L., a species of the Asteraceae family commonly known as cornflower or bachelor’s button, is an annual flowering plant native to Europe and the Middle East. *C. cyanus* grows up to 1–1.5 m tall annually, with strong stems, grayish, slightly hairy leaves, and small clusters of bright blue flowers. *C. cyanus* possesses several health-promoting effects and has been used for treating minor ocular inflammation [17] and as an antipruritic, antioxidant, anticancer, antitussive, astringent, mildly purgative, diuretic, and bitter tonic [18,19]. *C. cyanus* harbors a diverse array of potentially relevant bioactive constituents, including polysaccharides, polyphenols, flavonoids, and sesquiterpenes, documented in the literature [18]. These compounds have been linked to various pharmacological activities, such as anti-inflammatory, antioxidant, and anticancer effects, suggesting potential mechanisms by which *C. cyanus* might exert its muscle-protective properties [20,21]. Notably, while *C. cyanus* and its components have shown diverse pharmacological activities, their effects on skeletal muscle atrophy remain unreported. In this study, we aim to investigate the muscle-protective potential of *C. cyanus* water extract (CC) in both DEX-treated C2C12 myotubes and C57BL/6J mice. Furthermore, we delve into the underlying mechanisms of action, elucidating how specific active compounds within CC regulate muscle health and analyzing the impact of CC on gut microbiota composition.

## 2. Materials and Methods

### 2.1. Plant Materials and Extraction

The dried leaves of *C. cyanus* were collected in Cheongju, Republic of Korea, in November 2022 and purchased from a local market (Herbmaul, Cheongju, Chungcheongbuk-do, Republic of Korea). Prof. Dae-Sik Jang verified the plant identification (Kyung Hee University, Seoul, Republic of Korea). The leaves of *C. cyanus* (2.0 kg) underwent extraction using distilled water at room temperature, and the resulting extract was obtained through vacuum-assisted solvent removal. Subsequently, dried leaves of *C. cyanus* (2.0 kg), fragmented into pieces measuring 0.5–1 cm, underwent extraction using distilled water (20 L × 3 times) at room temperature for 24 h per cycle. The resulting residue (CC) was obtained after solvent removal under reduced pressure. The extract was stored at −20 °C until utilized in subsequent experiments.

### 2.2. Isolation and Identification of CC-Derived Compounds

CC was suspended in water and subsequently separated through partitioning using *n*-butyl alcohol (BuOH) to obtain the BuOH fraction (25 g) and water layer. The fraction soluble in BuOH was directly put onto a silica gel column using a gradient of MC (methylene chloride)-MeOH (20:1 to 0:20) to yield 32 subfractions (CCBu 1–32). Subfractions CCBu 3, 8, 25, and 31 were further purified using a Sephadex LH-20 column followed by preparative MPLC. Compound yields from each subfraction were as follows: compounds **7** (4.7 mg), **8** (2.8 mg), and **9** (1.5 mg) from CCBu-3; **1** (10.2 mg) from AEBu-8; **5** (3.6 mg) and **6** (2.7 mg) from CCBu-25; and **2** (2.6 mg), **3** (5.8 mg), and **4** (4.7 mg) from CCBu-31. Subfraction CCBu 3 (530 mg) was subjected to Sephadex LH-20 column using MeOH/H_2_O (1:10 to 100% MeOH) to collect compounds **7** (4.7 mg, 94.7%), **8** (2.8 mg, 92.5%), and **9** (1.5 mg, 92.1%). Subfraction CCBu 8 (310 mg) was applied to Sephadex LH-20 column using MeOH/H_2_O (2:10 to 100% MeOH) to obtain compound **1** (10.2 mg, 99.3%). Similarly, subfraction CCBu 25 (452 mg) and subfraction CCBu 31 (356 mg) were subjected to Sephadex LH-20 column using MeOH/H_2_O (3:10 to 100% MeOH) and further purified by preparative MPLC to yield compounds **2** (2.6 mg, 97.4%), **3** (5.8 mg, 94.6%), and **4** (4.7 mg, 91.3%) (Figure A1).

Graveobioside A (**5**): yellowish powder; ^1^H NMR (500 MHz, DMSO-*d*_6_) *δ* 7.46 (d, *J* = 8.3 Hz, 1H), 7.43 (d, *J* = 2.3 Hz, 1H), 6.91 (d, *J* = 8.3 Hz, 1H), 6.77 (d, *J* = 2.1 Hz, 1H), 6.76 (s, 2H), 6.44 (d, *J* = 2.2 Hz, 1H), 5.36 (d, *J* = 1.3 Hz, 1H), 5.19 (d, *J* = 7.2 Hz, 1H), 3.18–3.92 (m, 11H). ^13^C NMR (126 MHz, DMSO-*d*_6_) *δ* 181.88, 164.44, 162.65, 161.15, 156.89, 149.89, 145.75, 121.36, 119.17, 115.96, 113.53, 108.68, 105.33, 103.16, 99.29, 98.01, 94.63, 79.27, 76.98, 76.73, 76.02, 75.66, 73.97, 69.75, 64.20, 60.50. The details are provided in the Appendix A.

### 2.3. High-Performance Liquid Chromatography (HPLC) Profiling

The separation of analytes was achieved using HPLC on an Agilent 1200 series system (Agilent, St. Clara, CA, USA) equipped with a YMC Hydrosphere C18 column (YMC, Dinslaken, Germany). The column temperature was maintained at 30 °C throughout the analysis. To improve peak resolution, 0.1% formic acid was incorporated into both the acetonitrile and water mobile phases. A 10 µL injection volume was employed, and the flow rate was set to 1.0 mL/min. Detection was performed at 254 nm using UV absorbance.

### 2.4. Chemicals and Apparatus

One- and two-dimensional spectra of CC were obtained using Bruker Avance DRX Spectrometer (Billerica, MA, USA), with chemical shifts reported in ppm (δ). Mass spectrometry analyses were performed using electrospray ionization. Purification of the extract was achieved via open-column chromatography on silica gel (63–200 µm) and subsequent thin-layer chromatography using Merck Silica Gel 60 F254 and RP-18 F254 plates (Merck, Rahway, NJ, USA). Visualization of isolated compounds employed a standard protocol with 10% aqueous H_2_SO_4_ and heating for approximately 3 min. 

### 2.5. C2C12 Cell Differentiation and Cytotoxicity Assay

Following the protocol established in a previous publication [9], murine C2C12 myoblasts (CRL-1772, ATCC, Manassas, VA, USA) were maintained in Dulbecco’s modified Eagle’s medium (DMEM; Cytiva, Marlborough, MA, USA) supplemented with 10% heat-inactivated fetal bovine serum (Thermo Fisher Scientific, Waltham, MA, USA) and 1% penicillin/streptomycin (P/S; Cytiva) at 37 °C under a humidified atmosphere with 5% CO_2_. For differentiation, confluent cultures (70–80%) were switched to DMEM containing 2% horse serum (Thermo Fisher Scientific) and 1% P/S. Medium changes were performed every 3 days. To investigate the effects of CC and DEX, cells were treated with varying concentrations of CC (10, 20, or 40 μg/mL) for 24 h in the presence or absence of 100 μM DEX. DMSO served as the vehicle control. Cell viability was assessed using the EZ-Cytox kit (DoGenBio, Seoul, Republic of Korea) following the manufacturer’s protocol as previously described [22]. Briefly, differentiated C2C12 myotubes were seeded in 96-well plates at 10,000 cells per well and cultured in DMEM. EZ-Cytox reagent was then added, and the cells were incubated for an additional 2 h to determine viability. Optical density (OD) at 450 nm was measured using a microplate reader (Multiskan SkyHigh Microplate Spectrophotometer, Thermo Fisher Scientific).

### 2.6. Quantification of Myofiber

Differentiated C2C12 cells were fixed with 4% formaldehyde for 15 min and permeabilized with 0.4% Triton-X100 in Dulbecco’s phosphate-buffered saline (DPBS) for 10 min. Cells were then blocked with 10% donkey serum prepared in 0.1% Triton X-100 and 1% bovine serum albumin (BSA) in DPBS for 1 h, followed by overnight incubation with total myosin heavy chain (MHC) antibody (MF20, 1:100; Developmental Studies Hybridoma Bank). After washing three times with 0.1% Triton-X100/DPBS, cells were incubated with FITC-conjugated secondary antibody (1:100) alongside 1 μM Hoechst stain for 1 h at room temperature. Cells were again washed with 0.1% Triton-X100/DPBS and imaged using an Operetta High Content Imaging System (PerkinElmer Inc., Waltham, MA, USA). Myofiber area was quantified using appropriate image analysis software.

### 2.7. Detection of Reduced GSH Measurement

Differentiated C2C12 myotubes were washed with PBS, and cell lysates were collected using the Glutathione detection assay kit (Cell Signaling Technology, Danvers, MA, USA). After 60 min of incubation of samples lysates with assay buffer, reduced glutathione (GSH) was quantified using a microplate reader (Multiskan SkyHigh Microplate Spectrophotometer, Thermo Fisher Scientific) at excitation and emission wavelengths of 380 and 485 nm, respectively.

### 2.8. Mitotracker Assay

C2C12 cells were stained MitoTracker Green (Abcam, Cambridge, UK) at excitation/emission wavelengths of 485/525 nm, corresponding to probe excitation and fluorescence emission [9]. Mitochondrial activity was assessed using a microplate reader (Multiskan SkyHigh Microplate Spectrophotometer).

### 2.9. Determination of Cellular ATP Content

Cellular ATP content was measured using a CellTiter-Glo^®^ 2.0 Cell Viability Assay kit (Promega, Madison, WI, USA). Briefly, myotubes were washed after treatment with cold DPBS, homogenized in ATP sample buffer, and centrifuged at 13,000 rpm for 20 min. An amount of 50 μL of supernatant was added into each well of a 96-well plate, followed by the addition of ATP reaction mix solution at 25 °C for 30 min. The absorbance (OD 570 nm) was assessed by a microplate reader (Multiskan SkyHigh Microplate Spectrophotometer).

### 2.10. Measurement of ROS Production

Differentiated C2C12 myotubes were pretreated with various concentrations of CC for 24 h and stained with 20 μM DCFH-DA for 30 min. Cells were then treated with 10 μM DEX, and after 90 min, ROS production was measured at excitation/emission wavelengths of 485/530 nm using a microplate reader (Multiskan SkyHigh Microplate Spectrophotometer).

### 2.11. Computational Docking

For the docking of CC-derived compounds and reference compounds into the active sites of CB1 and CB2, we acquired atomic coordinates of CB1 and CB2 from the Protein Data Bank (PDB IDs: 5U09 and 6KPC, respectively). The target proteins were prepared using the protein preparation workflow in the Maestro 2023-02 release (Schrodinger LLC, New York, NY, USA). This involved standard procedures such as adding hydrogen atoms, assigning bond orders, eliminating water molecules, optimizing hydrogen bonds, and performing energy minimization using the OPLS4 force field. PROPKA was utilized to predict the ionization states of proteins. Optimization of hydrogen-bonding networks was carried out with ProtAssign in Maestro version 13.9.132. Ligands were prepared using LigPrep with Epik at pH 7.0 ± 2.0, considering protonation and tautomeric variations via the OPLS4 force field. Low-energy stereoisomers were generated and selected based on favorable 3D structures and correct chirality. The parameterized ligands were evaluated using induced-fit docking (IFD) for both CB1 and CB2.

### 2.12. Mouse Model of DEX-Induced Muscle Atrophy

Male C57BL/6J mice (10 weeks old) were purchased from DooYeol Biotech (Seoul, Republic of Korea) and housed under controlled conditions at 25 °C and 45% humidity under a 12 h light/12 h dark cycle. The mice received the AIN-76A standard diet (CA. 170481) (Envigo Teklad Diets, Indianapolis, IN, USA) and ad libitum water. After acclimatization, 60 mice were equally divided into six groups (n = 10 per group). The control group received 5% carboxymethyl cellulose (CMC) solution, while others received the same CMC solution containing oxymetholone (Sigma) (50 mg/kg/day), or CC at low (10 mg/kg/day), medium (50 mg/kg/day), or high (100 mg/kg/day) doses daily for one month. After 2 weeks, the groups were separated into control and DEX (n = 10/group). All DEX groups received intraperitoneal injections of DEX (25 mg/kg/day) for 14 days. The human equivalent dose was derived through the established body surface area approach [23]. The doses of 10, 50, and 100 mg/kg/day converted to human doses were 48.6, 243.2, and 486.4 mg/day for a 60 kg adult male. Body weights and grip strength were measured every other day. After the final injection and 16 h fasting, the quadriceps, gastrocnemius, soleus, and tibialis muscles were dissected, weighed, and frozen in liquid nitrogen. Muscle weight to body weight ratio was used as a primary measure of muscle atrophy.

### 2.13. Grip Strength and Exercise Performance Test

Grip strength was quantified using a dedicated grip force meter (Bioseb, Pinellas Park, FL, USA) by placing the mice on the grid and gently pulling them by the tail in the opposite direction until the forelimbs detached from the grid. Maximum intensity was recorded when the forelimbs were separated from the grid. Three trials were performed for each mouse, for a total of nine trials. The experimenters were blinded to the treatment conditions.

A Touchscreen Treadmill (Panlab, Harvard Apparatus, Holliston, MA, USA) was used for measuring exercise capacity and endurance. Prior to testing, mice were allowed to run for 15 min at 10 m/min. The treadmill test started at 10 m/min and then increased to 1 m/min until the mice were exhausted. A 20 V shock bar was positioned behind the treadmill to prevent the mouse from resting. Running distance and workload were calculated based on the time spent running and the corresponding speed. Exhaustion was defined as >10 s of continuous inactivity. The distance traveled was calculated by multiplying the travel time and the speed. 

### 2.14. Histological Analysis of Skeletal Muscle

Gastrocnemius muscle samples were fixed in 4% formalin overnight, then embedded in paraffin, and serially sliced to 4 µm for H&E staining. Quantitative analysis of muscle fiber area relative to total cross-sectional area was performed using ImageJ software (Version 1.54) [24]. The diameters of all muscle fibers in each view (100× objective, Zeiss, Oberkochen, Germany) were calculated using ImageJ software and marked as a percentage of the control.

### 2.15. Determination of Muscle ATP and Aconitase Contents

Tibialis muscle ATP and aconitase contents were measured using commercial assay kits (ATP Assay Kit and Aconitase Assay Kit, Abcam, Cambridge, UK) as per the manufacturer’s instructions. Briefly, homogenized muscle samples were subjected to centrifugation (13,000 rpm, 20 min), and 50 μL supernatants were added to microplate wells alongside the respective reaction mix solutions. After incubation at 25 °C for 30 min, absorbance (OD 570 nm) was measured (Multiskan SkyHigh Microplate Spectrophotometer) and normalized to tibialis muscle protein content.

### 2.16. RNA Extraction and Quantitative Reverse Transcription–Polymerase Chain Reaction (qRT-PCR)

Total mRNA was extracted from the tibialis muscle and cells using a HybridR RNA isolation kit (GeneAll, Seoul, Republic of Korea). The Reverse Transcription Premix was used to synthesize cDNA (ELPIS-Biotech, Daejeon, Republic of Korea). qRT-PCR was performed on a Light Cycler 480 Real-Time PCR System (Roche, Basel, Switzerland) using PowerUpTM SYBRTM Green Master Mix (Thermo Fisher Scientific) with gene-specific primers (summarized in Table 1). To account for potential variations, target mRNA expression was normalized against the housekeeping gene mouse β-actin. Relative gene expression was then calculated using the 2^−ΔΔCt^ method as previously described [22].

### 2.17. Western Blot Analysis

C2C12 myotubes (6-well plates, 1.6 × 10^5^ cells/well) were lysed in cold RIPA buffer (ThermoFisher Scientific) with phenylmethylsulfonyl fluoride and sodium orthovanadate (30 min on ice, 10 min vortexes). After centrifugation, equal protein amounts were loaded onto 10% SDS sulfate-polyacrylamide gels, separated by electrophoresis, transferred to PVDF membranes (Bio-Rad, Hercules, CA, USA), and blocked with 5% BSA (GenDEPOT). Membranes were incubated with specific primary antibodies overnight at 4 °C, washed 3 times with TBST, and incubated with anti-mouse (1:1000, Santa Cruz Biotechnology, Dallas, TX, USA) or anti-rabbit (1:5000, Santa Cruz Biotechnology) HRP-linked secondary antibodies for 2 h at 25 °C. Following TBST washes, protein bands were visualized using a chemiluminescence kit (ThermoFisher Scientific) and imaged on a LAS-4000 (Fujifilm, Minato, Japan). Band areas were quantified using ImageJ software as previously described [22].

### 2.18. 16S rRNA Gene Sequencing in Cecum Samples

Cecum content DNA was extracted using the QIAamp^®^ Fast DNA Stool Mini Kit (Qiagen, Hilden, Germany) after homogenization by bead beating. The V3–V4 hypervariable region of the 16S rRNA gene was amplified by PCR with barcoded primers and purified with AMPure XT beads (Beckman Coulter Genomics, Danvers, MA, USA). Sample quantification was performed using Qubit dsDNA reagent (Invitrogen, Carlsbad, CA, USA). Sequencing on a MiSeq platform (Illumina, San Diego, CA, USA) with paired-end 2 × 300 bp reads generated raw data analyzed using QIIME2-DADA2 for amplicon sequence variants (ASVs). Taxonomic classification against the SILVA database [25] was visualized at various levels (phylum, class, order, family, genus) using QIIME2. Alpha diversity (Shannon, Faith’s PD, and observed ASVs) and beta diversity (unweighted Unifrac and Bray–Curtis) were calculated with Phyloseq in R. Stack plots visualized abundance at each taxonomic level.

### 2.19. Statistical Analyses

R software (v.4.1.2) (R Core Team) was used to analyze the data. All data are presented as the mean ± standard error of the mean (SEM). One-way ANOVA followed by Tukey’s post hoc test (*p* < 0.05) was used to assess differences between groups. Spearman’s correlation was employed to investigate relationships between specific traits and microbial composition. Correlation coefficients and adjusted *p*-values (BH FDR procedure) were visualized using the ‘heatmap’ package. The differential abundance of genera was analyzed using ANCOM 2.1 and R software, with significance set at ANCOM W > 0.7. PERMANOVA (Vegan R package version 2.6-4) was employed to compare microbiota composition across groups.

## 3. Results

### 3.1. CC Demonstrates Protective Activity against DEX-Induced Atrophy in C2C12 Myotubes

DEX treatment significantly reduced myotube viability and diameter (32% and 30% decrease, respectively). However, CC treatment at 10 µg/mL significantly reversed these effects, increasing viability by 21% and diameter by 19.8% compared to the DEX group (Figure 1A–C) compared with that in the DEX group. CC did not exhibit cytotoxicity at concentrations up to 200 µg/mL (Figure A2A). Mechanistically, CC restored DEX-suppressed Akt phosphorylation and its downstream targets, p70S6K and 4E-BP1, while suppressing DEX-induced phosphorylation of FoxO3a and expression of MuRF1 (Figure 1D–F). Moreover, CC treatment reduced the protein and mRNA expression of Atrogin-1 (*Fbxo32*) and increased the expression of *Myd1*, *Myf6*, and *Myog*, which are MRFs (Figure 1F–I). These results suggest CC promotes protein synthesis, inhibits protein degradation, and stimulates myoblast differentiation via the AKT pathway.

### 3.2. CC Mitigates DEX-Induced Oxidative Stress and Boosts Mitochondrial Content in C2C12 Myotubes

DEX-induced muscle wasting is accompanied by increased oxidative stress, decreased glutathione activity, and mitochondrial loss and dysfunction [22]. DEX treatment increased ROS production (52%) and decreased glutathione (GSH) levels (13%) compared to control. However, CC treatment restored ROS (17% decrease) and GSH levels (12% increase) at 10 µg/mL to control levels and increased antioxidant gene expression (*Gpx1* and *Sod1*) (Figure 2A,D–F). Furthermore, CC reversed DEX-induced mitochondrial loss (136% increase at 10 µg/mL), as shown by MitoTracker Green staining, and restored ATP production to control levels at all concentrations tested (Figure 2B,C). Importantly, CC significantly upregulated mitochondrial biogenesis markers *Ppargc1a*, *Ucp3*, and *Tomm20* in DEX-treated myotubes (Figure 2G–I). These results suggest that CC may partially protect myofibers by inhibiting oxidative stress and improving mitochondrial content and function. 

### 3.3. Chemical Characterization of CC Reveals a Diverse Profile of Bioactive Phytochemicals

Next, we isolated nine known compounds (**1**–**9**) by processing with open-column chromatography to elucidate which of the secondary metabolites isolated from CC inhibit muscle wasting. These compounds were identified as protocatechuic acid (**1**) [26], neochlorogenic acid (**2**) [27], chlorogenic acid (**3**) [27], cryptochlorogenic acid (**4**) [27], graveobioside A (**5**) [28], apiin (**6**) [29], vanillic acid (**7**) [30], 4-hydroxybenzoic acid (**8**) [26,30], and uracil (**9**) [31] by comparing their physical and spectroscopic data to literature references (Figure 3 and Appendix A). Among these, compounds **1**, **3**, **5**, **6**, and **9** were the most abundant, with retention times of 4.3, 12.1, 17.1, 24.9, and 27.1 min, respectively (Figure 3A). 

### 3.4. Multiple Compounds Isolated from CC Exhibit Anti-Atrophic, Antioxidant, and ATP-Stimulatory Properties in DEX-Treated C2C12 Myotubes

Next, we sought to identify the active compounds from the CC-derived compounds that drive the myofiber protective effects. Cell cytotoxicity was assessed for eight of the isolated compounds (Figure A2B). Treatment (10 μM) of eight of the nine compounds isolated from CC, except for the nucleotide bases uracil (**9**), revealed that some compounds increased myofiber diameter and ATP and decreased ROS (Figure 4). Specifically, 5-*O*-caffeoylquinic acid (**3**) and 4-*O*-caffeoylquinic acid (**4**) increased muscle fibers and decreased ROS production (Figure 4A,B), while apiin (**6**), vanillic acid (**7**), and 4-hydroxybenzoic acid (**8**) simultaneously increased muscle fibers and ATP contents (Figure 4A,C). Notably, graveobioside A (**5**) exhibited the most significant effects, increasing myofiber diameter (12%), decreasing ROS (15%), and increasing ATP (8%) to control levels. These results suggest that multiple CC-derived compounds contribute to its protective effects against muscle atrophy and oxidative stress.

### 3.5. Graveobioside A Exerts Muscle-Protective Effects through Modulatory Action on Cannabinoid Receptors CB1 and CB2

Cannabinoid receptors CB1 and CB2 influence diverse physiological processes, including appetite, metabolism, and muscle function [32,33]. Previous studies suggest that CB1 inhibition or CB2 activation promotes muscle mass, strength, and exercise performance in mice [34,35,36]. We employed molecular docking to assess whether CC-isolated compounds modulate CB1 or CB2, potentially mitigating muscle atrophy. As shown in Table 2, among the eight compounds isolated from CC, 3-*O*-caffeoylquinic acid (compound **2**, −12.3 kcal/mol for CB1, −14.2 kcal/mol for CB2), graveobioside A (compound **5**, −17.7 kcal/mol for CB1, −19.1 kcal/mol for CB2) and apiin (compound **6**, −14.9 kcal/mol for CB1, −18.3 kcal/mol for CB2) commonly showed high affinity to CB1 and CB2. Notably, graveobioside A, with the strongest binding (Figure 5A), downregulated CB1 and upregulated CB2 protein and mRNA expression (Figure 5B). This modulation activated protein synthesis pathways (Figure 5C), suppressed protein degradation markers (Figure 5D–F), and enhanced mRNA expression of muscle cell differentiation (Figure 5G,H) and endurance-related markers (Figure 5I–L). These findings suggest that CC-derived compounds, particularly graveobioside A, improve muscle health through CB1 and CB2 regulation.

### 3.6. CC Ameliorates DEX-Induced Muscle Atrophy in a Murine Mode

To validate the in vivo muscle-protective effects of CC, we established a DEX-induced muscle atrophy model in mice (25 mg/kg body weight, intraperitoneal, 14 days; Figure A3). DEX significantly reduced body weight compared to controls (6.2% decrease; Figure 6A), indicating chronic toxicity. While CC did not fully recover weight loss, it completely reversed the DEX-induced grip strength decline (29.04% and 27.7% increase for low and high doses, respectively; Figure 6A,B). Histological analysis of gastrocnemius muscle revealed that DEX treatment decreased cross-sectional area (CSA) compared to controls, an effect reversed by all CC concentrations (*p* < 0.001; Figure 6C). Furthermore, CC significantly increased the mass of all four hindlimb muscles (quadriceps, gastrocnemius, tibialis, soleus) in DEX-treated mice (Figure 6D–G). Mechanistically, DEX downregulated myogenic regulatory factors (*Myog*, *Myod*, *Myf6*) and the structural muscle protein *Myh2* in the tibialis muscle, whereas CC treatment restored their expression (Figure 6H–K). These findings demonstrate that CC effectively protects against DEX-induced muscle wasting and dysfunction.

### 3.7. CC Enhances Exercise Capacity and Promotes Mitochondrial Function in Skeletal Muscle

DEX-induced muscle atrophy exhibits a characteristic depletion of type 2 glycolytic fibers, potentially impacting type 1 fibers with high mitochondrial content and fatigue resistance [37]. We investigated whether CC could improve exercise performance and mitochondrial biogenesis in DEX-treated mice. Treadmill testing revealed that DEX significantly reduced maximal speed, time to exhaustion, and distance compared to controls, while CC treatment significantly prolonged exercise performance, particularly at higher doses (Figure 7A–C). Additionally, high-dose CC reversed DEX-induced inhibition of ATP and aconitase production in the tibialis muscle (Figure 7D,E). Furthermore, CC upregulated *Ucp3*, *Nrf1*, and *Tfam* expression by increasing *Pparg1c* in DEX-treated mice (Figure 7F–I). These findings indicate that CC prevents DEX-induced muscle atrophy by inhibiting protein degradation pathways and enhancing mitochondrial function, thereby improving exercise capacity.

### 3.8. CC Modulates Gut Microbial Diversity and Composition, Reversing DEX-Induced Dysbiosis

Mounting evidence suggests the gut microbiota’s influence on muscle health may play a role in sarcopenia and cachexia [12]. Therefore, we performed 16S rRNA gene sequencing of mouse cecum content to assess the effect of CC on gut microbial diversity and composition. DEX treatment altered both phylum- and order-level bacterial composition (Figure A4A,B) and beta-diversity but not alpha-diversity indices (Shannon index, Faith’s PD, and ASV). Notably, DEX increased Bacteroidota and decreased Firmicutes compared to controls and exhibited significantly different beta-diversity distribution. In contrast, CC treatment at 10 and 50 mg/kg/day significantly increased alpha-diversity indices (Figure 8A,B and Figure A4C) and altered beta diversity compared to DEX (Adonis *p* < 0.001, Figure 8C and Figure A4D). ANCOM and Wilcoxon tests revealed that *Lachnospiraceae FCS020* and *Bifidobacterium pseudolongum*, taxa known to benefit muscle health, were significantly reduced in the DEX group and restored by CC10 treatment (Figure 8D and Figure A4E). Notably, *Lachnospiraceae FCS020* abundance increased in both low- and high-dose CC groups (Figure 8E,F, and Figure A4E–G). These findings suggest that CC promotes gut microbiota diversity and enrichment, counteracting DEX-induced dysbiosis and bringing it closer to the healthy control state.

### 3.9. Correlation Analysis Suggests a Potential Role for Clostridium Sensu Stricto 1 in Mediating the Muscle-Protective and Performance-Enhancing Effects of CC

Based on the ANCOM results, we investigated whether CC-altered bacterial taxa correlated with muscle-wasting and exercise performance traits. Association analyses between beta diversity and traits using Adonis revealed significant connections, suggesting gut microbiota composition plays a role in both (Figure 9A). Furthermore, genus-level correlation analyses identified the *Eubacterium brachy* group as positively associated with tibialis muscle mass and exercise performance, while *Clostridium sensu stricto 1* showed a strong positive correlation with grip strength and all four assessed muscle mass types (Figure 9B–F). These findings suggest that CC’s muscle-protective and performance-enhancing effects might be mediated, at least partially, by its ability to regulate the abundance of *Clostridium sensu stricto 1* within the gut microbiota.

## 4. Discussion

This study presents the first comprehensive investigation into the muscle-protective effects of CC and its bioactive constituents. We employed both in vitro (DEX-treated C2C12 myotubes) and in vivo (chronic DEX-induced muscle atrophy in mice) models. CC and its isolated compounds effectively countered DEX-induced muscle dysfunction. In C2C12 myotubes, CC enhanced cell viability, muscle fiber diameter, mitochondrial function, and myogenic differentiation while mitigating oxidative stress and protein degradation. Notably, graveobioside A, a CC-derived compound, emerged as a key player in this process, exerting its muscle-protective action through the modulation of cannabinoid receptors CB1 and CB2. In the in vivo model, CC treatment improved grip strength, exercise performance, and muscle fiber cross-sectional area. This was accompanied by enhanced mitochondrial function, increased muscle differentiation markers, and modulation of gut microbiota diversity.

FoxO3a phosphorylation has been identified as a crucial regulator of the ubiquitin–proteasome pathway, governing muscle protein degradation [38]. Glucocorticoid-induced muscle atrophy often involves preferential loss of fast-twitch type 2 glycolytic fibers [9]. These observations point towards PI3K/AKT pathway inhibition as a key factor in muscle breakdown under stress conditions. This involves dephosphorylation of FoxO3a, promoting its nuclear translocation and subsequent upregulation of E3 ubiquitin ligases Atrogin-1 and MuRF1 [39]. Building upon findings in previous studies, DEX treatment reduced phosphorylation of AKT and FoxO3a in both C2C12 myotubes and mouse models [40,41]. Conversely, CC counteracted these effects, suppressing proteolytic marker expression by inhibiting FoxO3a phosphorylation and stimulating AKT pathway activation. Additionally, muscle regulatory factors (MRFs) like MyoD, Myf5, Myog, and Myf6 play crucial roles in myogenesis by regulating MyHC, a major muscle protein [42]. Dysregulation of MRFs is implicated in both muscle atrophy and sarcopenia. MyoD and Myf5 initiate myoblast determination and activate Myf6 and Myog during further development [43,44]. Modulating the FoxO signaling pathway and MRF expression may be promising therapeutic targets for preventing or treating DEX-induced muscle atrophy. Our findings suggest that CC’s ability to modulate the FoxO signaling pathway and enhance MRF expression represents a promising therapeutic target for DEX-induced muscle atrophy.

Glutathione, a potent antioxidant molecule, plays a crucial role in protecting cells from damage caused by ROS. Studies have shown that muscle tissues from individuals with sarcopenia exhibit lower glutathione levels compared to healthy individuals [45]. This deficiency can trigger muscle damage and inflammation, leading to atrophy and extensive mitochondrial impairment [46]. Mitochondrial function and number are critical for maintaining muscle quality and function. Mitochondria generate ATP, the primary energy source for muscle contraction [47]. Reduced mitochondrial activity, observed in both sarcopenic individuals and mice, contributes to muscle weakness and wasting [48]. CC treatment effectively reversed DEX-induced muscle atrophy in mice, as evidenced by improved grip strength and treadmill performance. Moreover, it restored ATP and aconitase production in the tibialis muscle. Interestingly, CC stimulated Pparg1c mRNA expression, a transcription factor associated with mitochondrial function [49], and its target genes *Tfam* and *Nrf1*. Together, these findings suggest that CC protects muscle and myofibers from DEX-induced mitochondrial decline by increasing both glutathione content and *Pparg1c* activation.

In this study, we conducted an in-depth mechanistic investigation into the regulation of muscle wasting by testing eight isolated compounds from CC. These compounds significantly contributed to CC’s anti-atrophic activity by enhancing myofiber diameter and ATP content while mitigating oxidative stress. For example, 5-*O*-caffeoylquinic acid (**3**) and 4-*O*-caffeoylquinic acid (**4**) increased myofiber and inhibited ROS production. Supporting this, previous studies have documented the antioxidant properties of caffeoylquinic acid derivatives, demonstrating their ability to inhibit protein degradation and improve exercise performance [50,51,52]. These findings suggest that CC-derived compounds offer additional myofiber protective benefits beyond the collective action of CC. Graveobioside A emerged as a particularly active compound, significantly increasing myofiber diameter and ATP content and reducing oxidative stress. This potent effect likely stems from its binding affinity for cannabinoid receptors CB1 and CB2, influencing their expression. CB1 inhibition promotes mitochondrial metabolism, insulin sensitivity, and muscle growth, while CB2 activation enhances muscle metabolism and regeneration by upregulating MyoD and myogenin expression [53]. Consistent with this, studies involving the CB1 antagonist AM6475 or CB2 knockdown have demonstrated significant increases in muscle mass, grip strength, and physical endurance in mice [34,35,54]. Moreover, a mouse study with cancer-induced cachexia revealed that a selective CB2 agonist or overexpression of CB2 reduced muscle wasting and improved exercise performance [36]. In that regard, CB1 antagonists and CB2 agonists appear beneficial for muscle function in animal models, potentially mitigating muscle atrophy by inducing Akt and mTOR phosphorylation and influencing ubiquitin–proteasome system activity.

The gut–muscle axis, involving gut microbiota dysbiosis and sarcopenia, is a growing area of interest, particularly in older individuals [55]; however, its association remains unclear. Recent studies have shown that manipulating the gut microbiota through bacterial reduction, fecal transplantation, and supplementation can directly impact muscle phenotypes [11,12,13]. Consistent with these reports, a previous study observed that a probiotic suspension containing *Lactobacillus casei* LC122 and *Bifidobacterium longum* BL986 prevented age-related muscle loss and improved gut barrier function in 10-month-old C57BL/6 mice [56]. The gut microbiota potentially affects muscle performance by regulating protein synthesis/degradation, ATP production, lipid and glucose metabolism, inflammation, neuromuscular junctions, and mitochondrial function [57]. In the current study, CC treatment also impacted both alpha and beta diversity of the gut microbiota, along with increasing the enrichment of *Clostridium sensu stricto 1* at the genus level. While this opportunistic microorganism includes *Clostridium perfringens* and other real *Clostridium* species [58], its relationship with muscle health has not been extensively explored. Notably, in this study, *Clostridium sensu stricto 1* showed a positive correlation with muscle mass. Interestingly, *Clostridium sensu stricto 1*, identified as a producer of short-chain fatty acids (SCFAs), aligns with research demonstrating the muscle-protective benefits of dietary SCFA intake in older adults [59,60]. Butyrate, one of the representative SCFAs, promotes C2C12 myoblast proliferation [61] and alleviates muscle atrophy by enhancing gut barrier function and activating the PI3K/Akt/mTOR signaling cascade [62]. Therefore, further investigations are warranted to elucidate the potential protective role of *Clostridium sensu stricto 1* in the context of muscle atrophy.

## 5. Conclusions

This study demonstrates that CC exerts a multi-faceted protective effect against skeletal muscle atrophy through distinct mechanisms. By mitigating protein degradation, boosting protein synthesis, and enhancing mitochondrial biogenesis, CC directly improves muscle function. Additionally, CC modulates gut microbiota, promoting a composition beneficial for muscle health. Notably, our integrated in silico and in vitro analyses reveal that specific bioactive compounds isolated from CC act by regulating CB1 and CB2 receptors, offering a potential mechanism for their protective effects against DEX-induced muscle atrophy. These findings highlight the promise of CC and its isolated bioactive components for promoting muscle health and preventing muscle wasting in conditions like DEX-induced atrophy, acting through distinct yet complementary modes of action.

## Figures and Tables

**Figure 1 nutrients-16-01130-f001:**
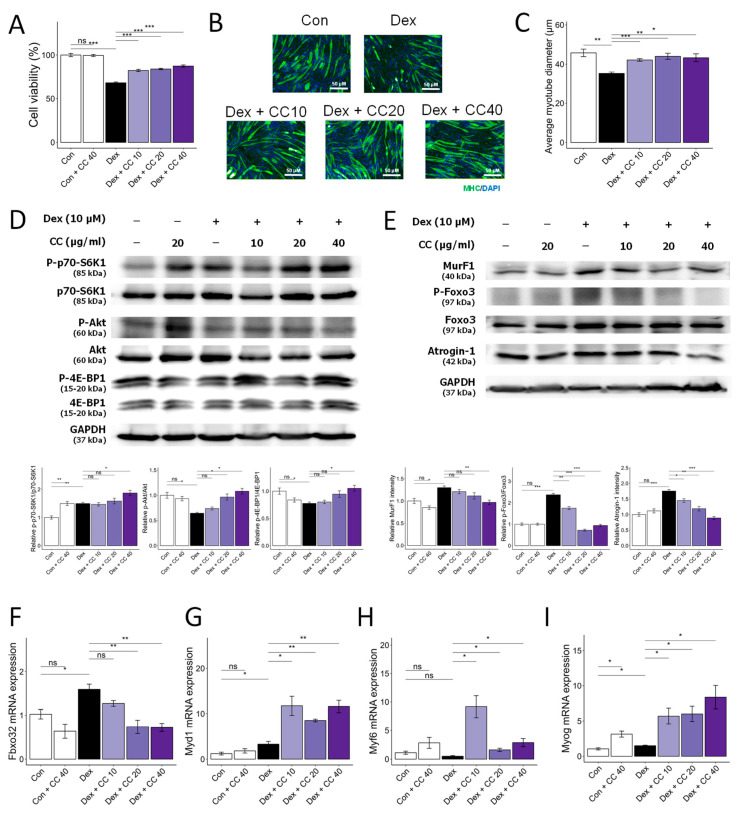
*Centaurea cyanus* extract (CC) protects dexamethasone (Dex)-induced muscle atrophy in C2C12 myotubes. (**A**) Cell viability of C2C12 myotubes treated with various concentrations of CC (10, 20, 40 µg/mL) in the presence or absence of Dex (100 µM) for 24 h. (**B**,**C**) Representative images of C2C12 cells stained with myosin heavy chain antibody (**B**) and average myotube diameter (**C**) quantified by ImageJ software. (**D**) Protein expression levels of phospho-p70-S6K1, p70-S6K1, phospho-Akt, Akt, phosphor-4E-BP1, and 4E-BP1. GAPDH was used as a housekeeping gene control, and intensity was quantified by ImageJ software. The bar graph shows the ratio of the phosphorylated form to the total form for each protein. (**E**) Protein expression levels of MurF1, phospho-FoxO3a, FoxO3a, and Atrogin-1. GAPDH was used as a housekeeping gene control, and intensity was quantified by ImageJ software. The bar graph shows the relative intensity of each protein. (**F**–**I**) mRNA expression levels of muscle cell differentiation markers, including *Fbxo32*, *Myod1*, *Myf6*, and *Myog*. β-actin was used as a housekeeping gene control. Data are shown as mean ± SE, n = 3–4. Tukey’s post hoc multiple comparisons test was performed. * *p* < 0.05, ** *p* < 0.01, and *** *p* < 0.005; CC or control vs. Dex; ns (not significant) *p* > 0.05. Dex, dexamethasone; CC, *Centaurea cyanus* extract.

**Figure 2 nutrients-16-01130-f002:**
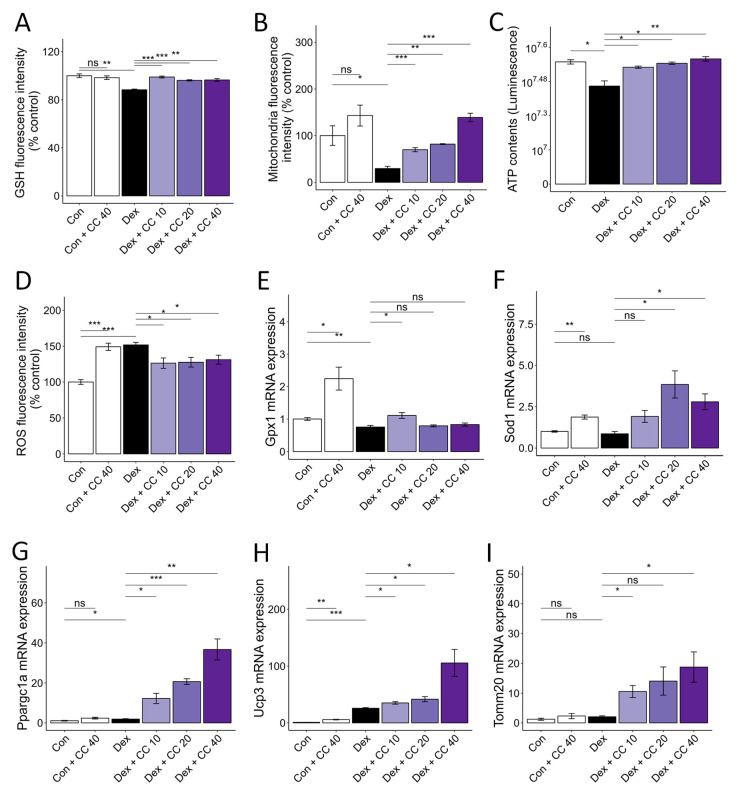
CC reverses Dex-induced oxidative stress and mitochondrial dysfunction in C2C12 myotubes. (**A**–**D**) Glutathione content (**A**), mitochondrial content (**B**), ATP concentration (**C**), and reactive oxygen species level (**D**) in Dex-treated C2C12 myotubes. (**E**,**F**) mRA expression levels of *Gpx1* and *Sod1*. (**G**–**I**) mRNA expression levels of *Pparg1a*, *Ucp3*, and *Tomm20*. β-actin was used as a loading control. Data are shown as mean ± SE, n = 3–4. * *p* < 0.05, ** *p* < 0.01, and *** *p* < 0.005, ns (not significant) *p* > 0.05.

**Figure 3 nutrients-16-01130-f003:**
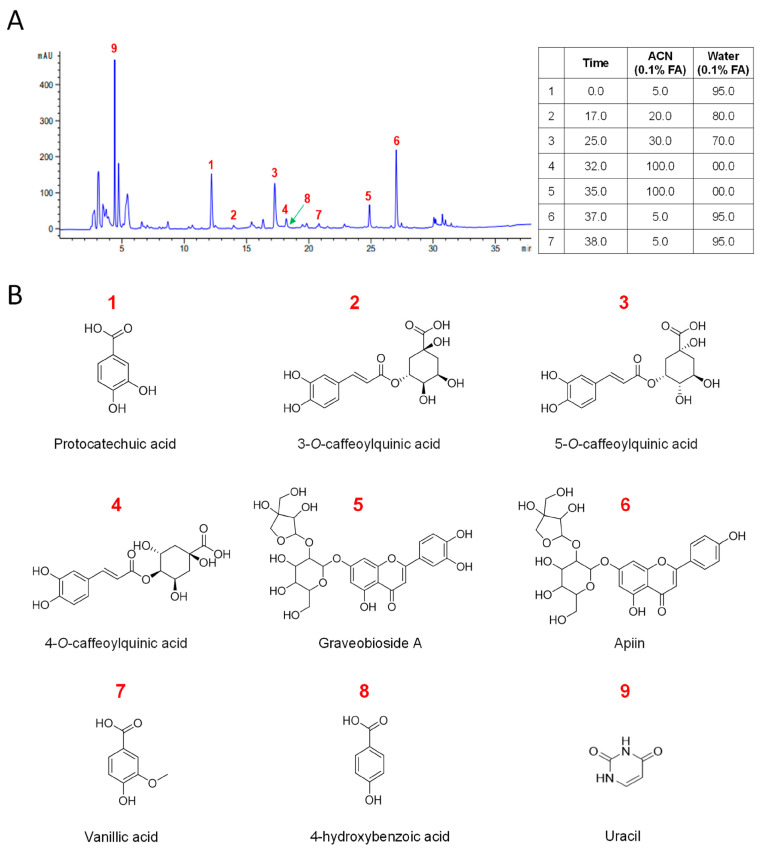
Chemical profiling of CC water extract and identification of its major compounds. (**A**) The chromatograms of the water extract of *Centaurea cyanus.* (**B**) Structures of isolated compounds from CC.

**Figure 4 nutrients-16-01130-f004:**
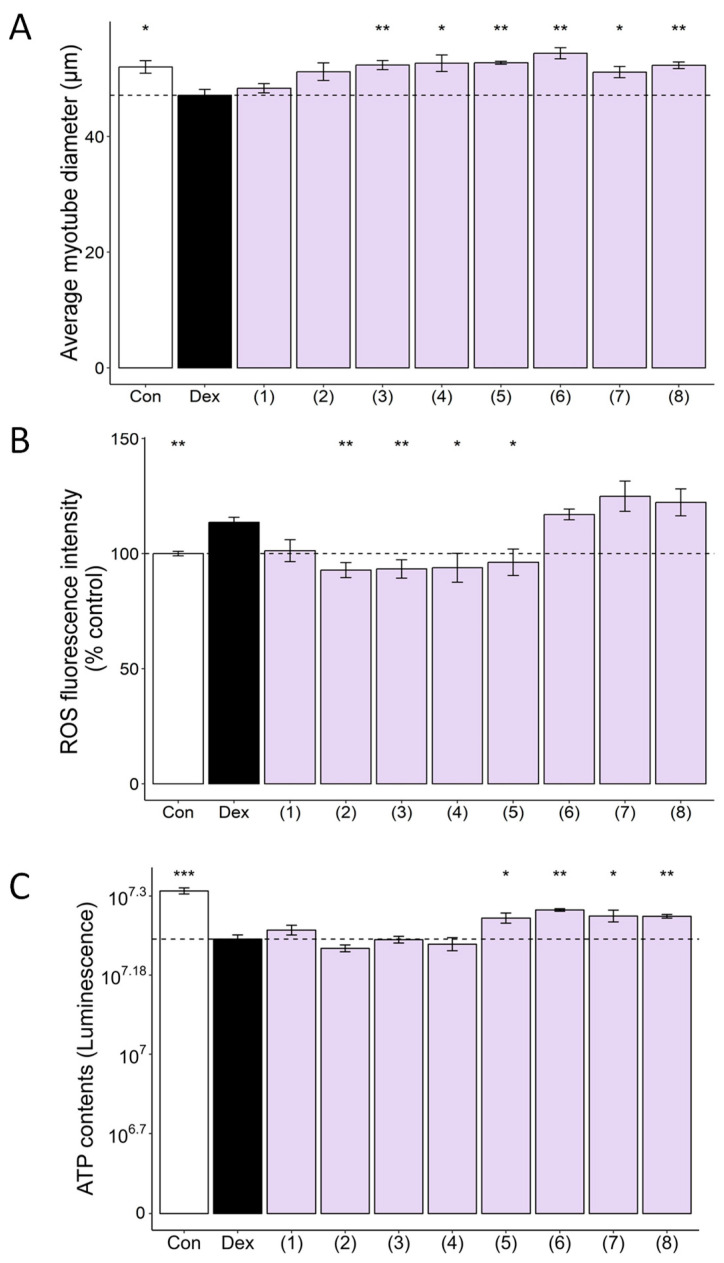
CC-isolated compounds ameliorate Dex-induced muscle atrophy, oxidative stress, and mitochondrial dysfunction in C2C12 myotubes. CC-isolated compounds are protocatechuic acid (**1**), neochlorogenic acid (**2**), chlorogenic acid (**3**), cryptochlorogenic acid (**4**), graveobioside A (**5**), apiin (**6**), vanillic acid (**7**), and 4-hydroxybenzoic acid (**8**). (**A**) Average myotube diameter quantified by ImageJ software. (**B**) The effect of CC-derived compounds on ROS generation in Dex-treated C2C12 myotubes, as assessed by DCFDA staining. (**C**) The effect of CC on ATP concentration in Dex-treated C2C12 myotubes. Data are shown as mean ± SE, n = 3–4. * *p* < 0.05, ** *p* < 0.01, and *** *p* < 0.005.

**Figure 5 nutrients-16-01130-f005:**
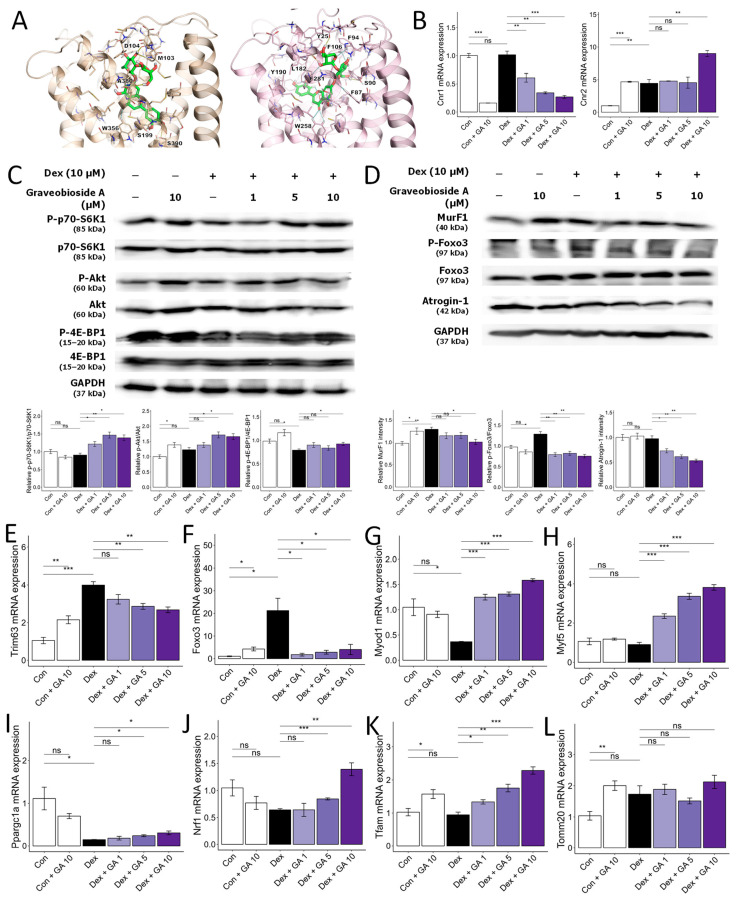
Graveobioside A inhibits Dex-induced muscle atrophy by regulating CB1 and CB2 in C2C12 myotubes. (**A**) Induced-fit docking poses of graveobioside A (green) in CB1 (light orange) and CB2 (pink) are presented in the panel. Residues located within 5 Å of the ligand are displayed as sticks. Hydrogen bonds and aromatic hydrogen bonds are denoted by yellow dashed lines and cyan dashed lines, respectively. The residues engaged in these interactions are labeled accordingly. (**B**) Effect of graveobioside A (GA) on mRNA expression levels of CB1 and CB2. β-actin was used as a housekeeping gene control. (**C**) Protein expression levels of phospho-p70-S6K1, p70-S6K1, phospho-Akt, Akt, phospho-4E-BP1, and 4E-BP1. GAPDH was used as a housekeeping gene control, and intensity was quantified by ImageJ software. The bar graph shows the ratio of the phosphorylated form to the total form for each protein. (**D**) Protein expression levels of MurF1, phospho-FoxO3a, FoxO3a, and Atrogin-1. GAPDH was used as a housekeeping gene control, and intensity was quantified by ImageJ software. The bar graph shows the relative intensity of each protein. (**E**–**L**) mRNA expression levels of *Trim63*, *Foxo*, *Myod1*, *Myf5*, *Ppargc1a*, *Nrf1*, *Tfam*, and *Tomm20*. β-actin was used as a housekeeping gene control. Data are shown as mean ± SE, n = 3–4. * *p* < 0.05, ** *p* < 0.01, and *** *p* < 0.005, ns (not significant) *p* > 0.05.

**Figure 6 nutrients-16-01130-f006:**
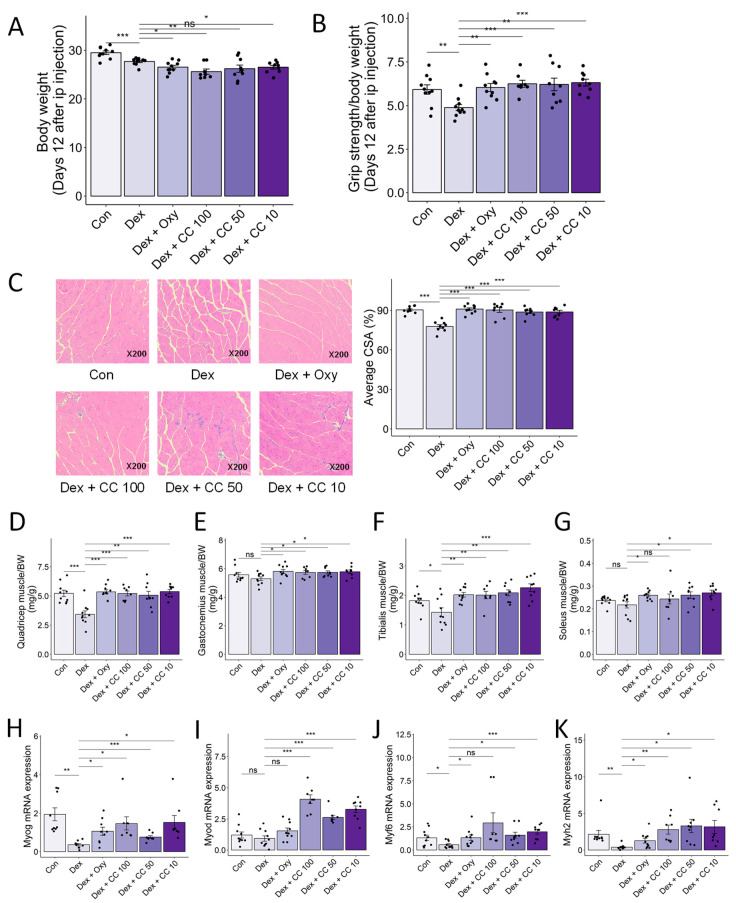
CC protects Dex-induced muscle weakness. (**A**) Body weight. (**B**) Grip test strength measured before sacrifice. (**C**) Gastrocnemius (GA) muscles were stained with H&E staining, and representative images were shown. The average cross-sectional area (CSA) of GA muscle fiber was quantified by ImageJ. (**D**–**G**) Quadriceps (**D**), gastrocnemius (**E**), tibialis (**F**), and soleus (**G**) muscle masses relative to total body weight. (**H**–**K**) mRNA expression levels of *Myog*, *Myod1*, *Myf6*, and *Myh2* in tibialis muscle. β-actin was used as a housekeeping gene control. Data are shown as mean ± SE, n = 8–10. The black dots on the bar graph represent the data values for each mouse on the x-axis. * *p* < 0.05, ** *p* < 0.01, and *** *p* < 0.005, ns (not significant) *p* > 0.05.

**Figure 7 nutrients-16-01130-f007:**
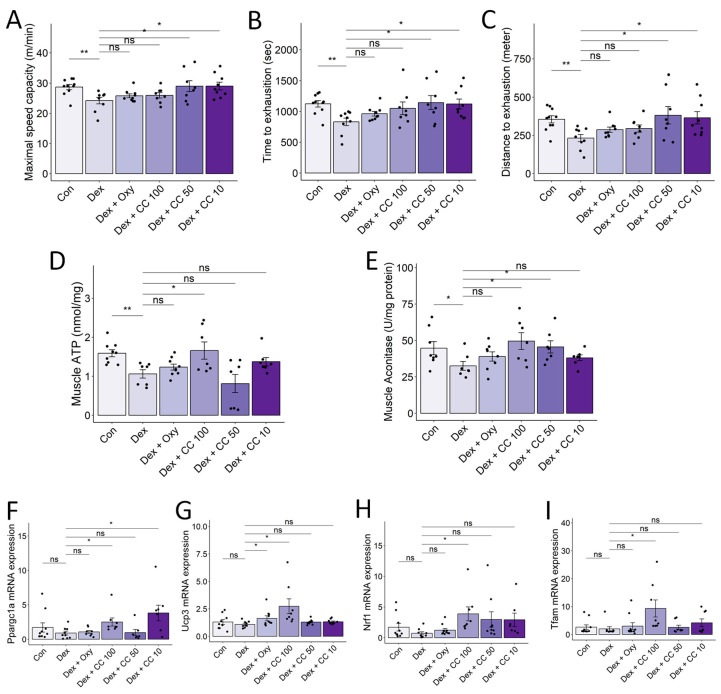
CC increases exercise capacity and mitochondrial functions in skeletal muscle. (**A**–**C**) Maximal speed (**A**) and time (**B**) and distance (**C**) to exhaustion were evaluated. (**D**,**E**) ATP and aconitase activity in the tibialis muscle was normalized to protein content in the homogenate. (**F**–**I**) mRNA expression levels of *Pparg1a*, *Ucp3*, *Nrf1*, and *Tfam*. β-actin was used as a housekeeping gene control. Data are shown as mean ± SE, n = 8–10. The black dots on the bar graph represent the data values for each mouse on the x-axis. * *p* < 0.05, ** *p* < 0.01, ns (not significant) *p* > 0.05.

**Figure 8 nutrients-16-01130-f008:**
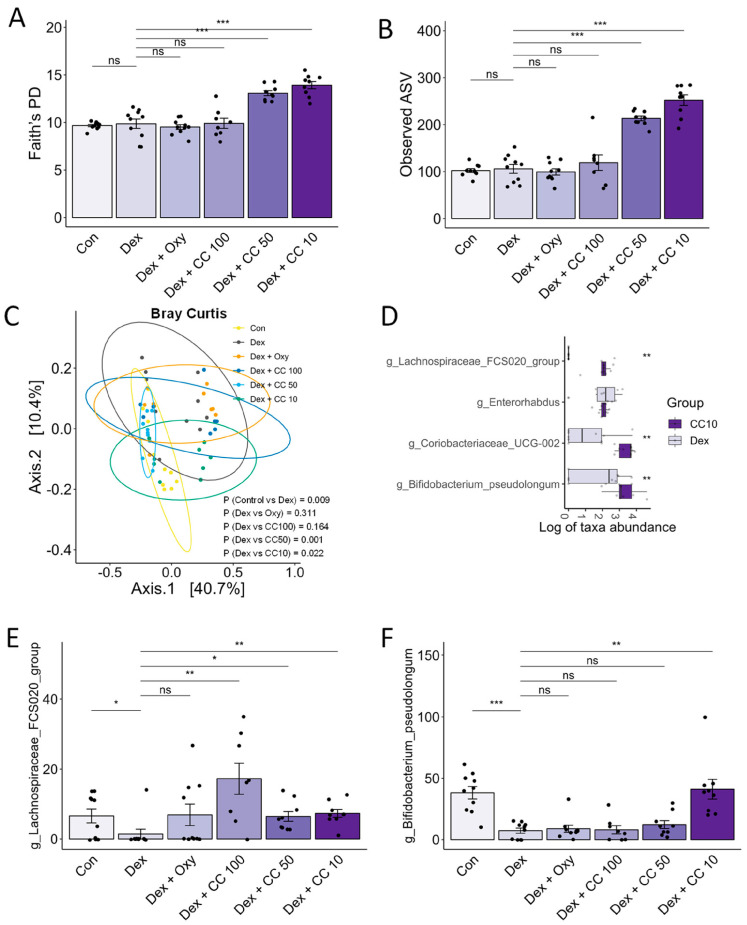
CC modulates gut microbial diversity and bacterial taxa. (**A**–**C**) Alpha- and beta-diversity indices, Faith’s PD (**A**), observed ASV (**B**), and Bray–Curtis index (**C**). (**D**) Genera showing significantly differential abundance between the Dex and CC10 groups. (**E**) Abundances of *Lachnospiraceae FCS020* and (**F**) *Bifidobacterium pseudolongum.* Data are shown as mean ± SE, n = 8–10. The black dots on the bar graph represent the data values for each mouse on the x-axis. * *p* < 0.05, ** *p* < 0.01, and *** *p* < 0.005, ns (not significant) *p* > 0.05.

**Figure 9 nutrients-16-01130-f009:**
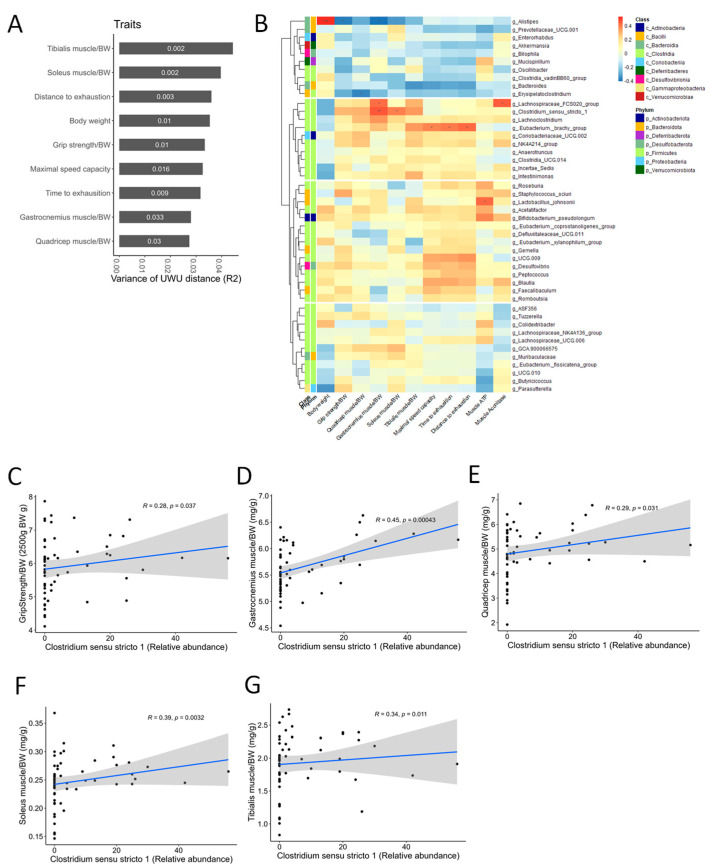
The correlation analysis between the relative abundance of *Clostridium sensu stricto* 1 and sarcopenia-related markers. (**A**) Association between the variance of the gut microbial beta-diversity (Bray–Curtis, BC) distance and sarcopenia-related traits. The numbers in the bar graph are *p*-values. (**B**) The heat map shows the correlations between the abundance of microbial genera and sarcopenia-related markers. In taxonomic classification, the class and phylum level to which each genus belongs are denoted with different colors. The *p*-values were adjusted using the Benjamini–Hochberg (BH) FDR procedure. “***” *p* < 0.001, “**” *p* < 0.01, “*” *p* < 0.05, “.” *p* < 0.10. (**C**–**G**) Spearman correlations between the relative abundance of *Clostridium sensu stricto 1* and sarcopenia-related markers (grip strength and four types of muscle mass). The black dots represent the data points for each mouse. The blue line represents the first-order correlation function. The gray area represents the confidence interval.

**Table 1 nutrients-16-01130-t001:** qRT-PCR mouse primer sequences (5′-3′).

Species	Target Gene	Direction	Primer Sequence (5′-3′)	Gene ID
Mouse	*Fbxo32*	Forward	AACCCTTGGGCTTTGGGTTT	NM_026346.3
		Reverse	GGACTTAAGCCCGTGCAGAT	
	*Myod1*	Forward	CATAGACTTGACAGGCCCCG	NM_010866.2
		Reverse	CGGGTCCAGGTCCTCAAAAA	
	*Myf6*	Forward	ACAGATCGTCGGAAAGCAGC	NM_008657.3
		Reverse	CACTCCGCAGAATCTCCACC	
	*Myog*	Forward	AGCTATCCGGTTCCAAAGCC	NM_031189.2
		Reverse	GCACAGGAGACCTTGGTCAG	
	*Myh2*	Forward	AGCGAAGAGTAAGGCTGTCC	NM_001039545.2
		Reverse	AGGCGCATGACCAAAGGTT	
	*Ppargc1a*	Forward	GTTGCCTGCATGAGTGTGTG	NM_008904.3
		Reverse	CACATGTCCCAAGCCATCCA	
	*Ucp3*	Forward	GTTTTGCGGACCTCCTCACT	NM_009464.3
		Reverse	CTCTGTGCGCACCATAGTCA	
	*Tomm20*	Forward	TGTGCGGTGTGTTGTCTGTT	NM_024214.2
		Reverse	TAAGTGCCCAGAGCACAGGA	
	*Nrf1*	Forward	CCCGTGTTCCTTTGTGGTGA	NM_001410231.1
		Reverse	ATTCCATGCTCTGCTGCTGG	
	*Tfam*	Forward	GGGAATGTGGAGCGTGCTAA	NM_009360.4
		Reverse	TGATAGACGAGGGGATGCGA	
	*Gpx1*	Forward	AGTCCACCGTGTATGCCTTC	NM_001329527.1
		Reverse	CCTCAGAGAGACGCGACATT	
	*Sod1*	Forward	GGGAAGCATGGCGATGAAAG	NM_011434.2
		Reverse	GCCTTCTGCTCGAAGTGGAT	
	*Actb*	Forward	CATTGCTGACAGGATGCAGAAGG	NM_007393.5
		Reverse	TGCTGGAAGGTGGACAGTGAGG	

**Table 2 nutrients-16-01130-t002:** Induced-fit docking scores of CB1 and CB2 reference compounds and CC-isolated compounds.

Compounds	Docking Score (kcal mol^−1^)	
	CB1	CB2
Rimonabant (CB1 antagonist)	−13.215	−12.170
AM251 (CB1 antagonist)	−12.771	−12.579
GW-405833 (CB2 agonist)	−12.367	−12.283
CB65 (CB2 agonist)	−12.711	−11.491
Protocatechuic acid (**1**)	−7.006	−8.002
3-*O*-caffeoylquinic acid (**2**)	−12.293	−14.220
5-*O*-caffeoylquinic acid (**3**)	−11.642	−13.024
4-*O*-caffeoylquinic acid (**4**)	−12.845	−13.020
Graveobioside A (**5**)	−17.714	−19.098
Apiin (**6**)	−14.865	−18.306
Vanillic acid (**7**)	−6.648	−7.291
4-hydroxybenzoic acid (**8**)	−5.981	−6.823

## Data Availability

Data are contained within the article and Appendix A.

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
