# Peer review of "Cornflower Extract and Its Active Components Alleviate Dexamethasone-Induced Muscle Wasting by Targeting Cannabinoid Receptors and Modulating Gut Microbiota"

_nutrients, 2024, doi:10.3390/nu16081130_

Round 1

Reviewer 1 Report

Comments and Suggestions for Authors

Interesting work on research on preventing the loss of body weight and muscle strength by substances contained in the water extract of cornflower. This extract alleviated oxidative stress, promoted muscle growth, and increased ATP production in C2C12 myotubes. It also reduced protein degradation markers, increased mitochondrial content, and activated protein synthesis signaling. We assessed grip strength, exercise performance, and modulation of muscle gene expression related to differentiation, protein turnover, and exercise performance. This extract influenced the microbial diversity of the intestines. These studies may have practical applications in the prevention and treatment of muscle atrophy, and thus in the treatment of diseases in which such symptoms occur.

The materials and methods are thorough and precise. Statistical analysis includes well-selected tests. Statistically significant differences were found. Clean charts. In brake charts marked with different letters, differences (statistically significant differences can be marked with different links or capital letters above the bars) between trials. References include well-selected literature.

Author Response

Reviewer #1:

Interesting work on research on preventing the loss of body weight and muscle strength by substances contained in the water extract of cornflower. This extract alleviated oxidative stress, promoted muscle growth, and increased ATP production in C2C12 myotubes. It also reduced protein degradation markers, increased mitochondrial content, and activated protein synthesis signaling. We assessed grip strength, exercise performance, and modulation of muscle gene expression related to differentiation, protein turnover, and exercise performance. This extract influenced the microbial diversity of the intestines. These studies may have practical applications in the prevention and treatment of muscle atrophy, and thus in the treatment of diseases in which such symptoms occur.

The materials and methods are thorough and precise. Statistical analysis includes well-selected tests. Statistically significant differences were found. Clean charts. In brake charts marked with different letters, differences (statistically significant differences can be marked with different links or capital letters above the bars) between trials. References include well-selected literature.

-> Thank you very much for your insightful review and positive assessment of our research on the water extract of cornflower (CC) and its potential to prevent muscle loss and weakness. We are particularly encouraged by your appreciation for the thoroughness of our methods, the selection of statistical tests, the clarity of our charts, and the relevance of our cited literature. We are especially grateful for your specific comments regarding the potential practical applications of our findings. As you mentioned, CC's ability to alleviate oxidative stress, promote muscle growth, and influence gene expression related to muscle function suggests its potential role in preventing and treating muscle atrophy associated with various diseases. We believe our research lays the groundwork for further investigation into the therapeutic potential of CC in this context. In regards to your suggestion on differentiating statistically significant differences in the bar charts, we have increased the font size of the figures and the size of the significance symbols above the bars for improved clarity in the revised version of the manuscript. Thank you again for your valuable feedback. We appreciate your contribution to strengthening our research.

Reviewer 2 Report

Comments and Suggestions for Authors

This is a good work. Some suggestions were as follows:

1. In the section of Introduction, a brief review on cornflower, particularly health-benefiting ingredients should be detailed.

2. In the section of 2.1. Plant Materials and Extraction, the preparation of cornflower water extract (CC) should be detailed as well as the phytochemical analysis, including carbohydrates, phenols, flavonoids, and the like.

3. In the Section of 2.2. Isolation and Identification of CC-Derived Compounds, a flow chart of isolation would be helpful to better understand the procedure. Moreover, the weight and purity of each compound should be replenished.

4. Generally, polysaccharides and other water-soluble constituents could enrich in water extract, why did the authors not consider the contribution of these ingredients to the activity of CC?

5. Unlike monomer compounds, the activity of extracts is often contributed by the synergistic action of multiple components, unless the content of investigated compound is predominant at least > 50%. Therefore, in my view, for extracts, molecular docking is of little significance, and activity-directed isolation is more conducive to the discovery of real activity-contributing components or monomer compounds.

Author Response

Reviewer #2:

This is a good work. Some suggestions were as follows:

  1. In the section of Introduction, a brief review on cornflower, particularly health-benefiting ingredients should be detailed.

-> In accordance with the reviewer's suggestion, we have incorporated a detailed description of Centaurea cyanus, including its physiological activities and chemical constituents, into the introduction section to provide a comprehensive context for our research.

Page 4, lines 81-95: Centaurea cyanus L., a species of the Asteraceae family commonly known as cornflower or bachelor’s button, is an annual flowering plant native to Europe and the middle east. C. cyanus grows up to 1–1.5 m tall annually, with strong stems, grayish, slightly hairy leaves, and small clusters of bright blue flowers. C. cyanus possess several health-promoting effects, and has been used as minor ocular inflammation [17] an antipruritic, antioxidant, anticancer, antitussive, astringent, mildly purgative, diuretic, and bitter tonic [18, 19]. C. cyanus harbors a diverse array of potentially relevant bioactive constituents, including polysaccharides, polyphenols, flavonoids, and sesquiterpenes, documented in the literature [18]. These compounds have been linked to various pharmacological activities, such as anti-inflammatory, antioxidant, and anticancer effects, suggesting potential mechanisms by which C. cyanus might exert its muscle-protective properties [20, 21] . Notably, while C. cyanus and its components have shown diverse pharmacological activities, their effects on skeletal muscle atrophy remain unreported. In this study, we aim to investigate the muscle-protective potential of C. cyanus water extract (CC) in both DEX-treated C2C12 myotubes and C57BL/6J mice.

  1. In the section of 2.1. Plant Materials and Extraction, the preparation of cornflower water extract (CC) should be detailed as well as the phytochemical analysis, including carbohydrates, phenols, flavonoids, and the like.

-> Thank you for your valuable feedback. We have addressed your suggestions as follows:

  • Improved Details of CC Extraction: We have incorporated a more detailed description of the cornflower water extract (CC) preparation process in Section 2.1 of the revised manuscript (page 5, lines 105-109). This includes specifying the degree of leaf crushing, solvent volume, extraction cycles, extraction time, and extract storage conditions.

Page 5, lines 105-109: Subsequently, dried leaves of C. cyanus (2.0 kg), fragmented into pieces measuring 0.5–1 cm, underwent extraction using distilled water (20 L × 3 times) at room temperature for 24 hours per cycle. The resulting residue (CC) was obtained after solvent removal under reduced pressure. The extract was stored at −20°C until utilized in subsequent experiments.

  • Phytochemical Analysis: While we appreciate your suggestion to include data on various components like carbohydrates, phenols, and flavonoids, our primary focus in this study was on identifying individual compounds rather than conducting a comprehensive analysis of their total content. We believe this approach was more aligned with the specific objectives of our research.
  1. In the Section of 2.2. Isolation and Identification of CC-Derived Compounds, a flow chart of isolation would be helpful to better understand the procedure. Moreover, the weight and purity of each compound should be replenished.

-> Thank you for your valuable feedback on Section 2.2 of our manuscript. We agree that a flow chart would enhance clarity regarding the isolation procedure of CC-derived compounds. We have included a detailed flow chart outlining the isolation process (Fig. S1). We also recognize the importance of reporting the weight and purity of each isolated compound. In our revised manuscript, we have added this information in the method section (page 6, lines 118), providing accurate data on the weight and purity of each compound obtained through our isolation process.

Page 6, lines 118-132: Subfraction CCBu 3 (530 mg) was subjected to Sephadex LH-20 column using MeOH/H2O (1:10 to 100% MeOH) to collect compounds 7 (4.7 mg, 94.7%), 8 (2.8 mg, 92.5%), and 9 (1.5 mg, 92.1%).  Subfraction CCBu 8 (310 mg) was applied to Sephadex LH-20 column using MeOH/H2O (2:10 to 100% MeOH) to obtain compound 1 (10.2 mg, 99.3%). Similarly, subfraction CCBu 25 (452 mg) and subfraction CCBu 31 (356 mg) subjected to Sephadex LH-20 column using MeOH/H2O (3:10 to 100% MeOH) and further purified by preparative MPLC to yield compounds 2 (2.6 mg, 97.4%), 3 (5.8 mg, 94.6%), and 4 (4.7 mg, 91.3%) (Figure xx). Graveobioside A (5): yellowish powder; 1H NMR (500 MHz, DMSO-d6) δ 7.46 (d, J = 8.3 Hz, 1H), 7.43 (d, J = 2.3 Hz, 1H), 6.91 (d, J = 8.3 Hz, 1H), 6.77 (d, J = 2.1 Hz, 1H), 6.76 (s, 2H), 6.44 (d, J = 2.2 Hz, 1H), 5.36 (d, J = 1.3 Hz, 1H), 5.19 (d, J = 7.2 Hz, 1H), 3.18  ̶ 3.92 (m, 11H). 13C NMR (126 MHz, DMSO-d6) δ 181.88, 164.44, 162.65, 161.15, 156.89, 149.89, 145.75, 121.36, 119.17, 115.96, 113.53, 108.68, 105.33, 103.16, 99.29, 98.01, 94.63, 79.27, 76.98, 76.73, 76.02, 75.66, 73.97, 69.75, 64.20, 60.50. The details are provided in Supplementary file.

Supplementary Figure 1. Flow chart for isolation and identification of CC-derived compounds

  1. Generally, polysaccharides and other water-soluble constituents could enrich in water extract, why did the authors not consider the contribution of these ingredients to the activity of CC?

-> We appreciate the reviewer's thoughtful question regarding the potential contribution of polysaccharides and other water-soluble constituents to cornflower water extract (CC) activity. Our study deliberately focused on isolating and analyzing the bioactivity of lipophilic compounds, which aligns with our research objectives. While acknowledging that water-soluble components like polysaccharides could influence overall activity, we aimed to provide specific insights into the lipophilic fraction's potential contributions. Future investigations encompassing both lipophilic and water-soluble fractions, as suggested by the reviewer, could offer a more holistic understanding of CC's overall activity. We value this feedback and will consider incorporating this broader perspective in future research endeavors.

  1. Unlike monomer compounds, the activity of extracts is often contributed by the synergistic action of multiple components, unless the content of investigated compound is predominant at least > 50%. Therefore, in my view, for extracts, molecular docking is of little significance, and activity-directed isolation is more conducive to the discovery of real activity-contributing components or monomer compounds.

-> We appreciate your valuable perspective regarding the limitations of molecular docking for complex extracts. We acknowledge that the activity of extracts often stems from the synergistic action of multiple components, particularly when the concentration of a single constituent is below 50%. In such cases, activity-directed isolation undoubtedly plays a crucial role in isolating and identifying the key active components. In our study, we employed molecular docking alongside activity-directed isolation. While we utilized docking to explore potential interactions between individual compounds and the target molecule, gaining insights into their potential pharmacological activities, we recognize the critical role of activity-directed isolation in identifying the true contributors to the observed activity. Therefore, we believe the combined approach provides a more comprehensive understanding of the extract's mechanism of action. We will carefully consider your feedback and re-evaluate our approach in future studies, potentially prioritizing activity-directed isolation when dealing with complex extracts with low concentrations of individual components. Thank you for your valuable insight, which will undoubtedly strengthen our future research endeavors.

Reviewer 3 Report

Comments and Suggestions for Authors

1. Considering that dexamethasone is the specific glucocorticoid employed for inducing muscle atrophy in this study, it is recommended to replace "glucocorticoid" with "dexamethasone" in the title.

2. The Introduction fails to provide the necessary background information on Centaurea cyanus.

3. The full name of the abbreviation CC should be indicated when it first appears in the main text.

4. The description of the method of isolation and identification of compounds derived from CC is too simple, and should be described in detail.

5. The number 5 in Line 243:10 to the fifth power should be superscript.

6. Line 255: "16.s" should be changed to "16s"

7. Figure 1E: The results of p-Foxo3 should be updated with an unblemished one.

8. The titles and significance labels of the bar chart's horizontal and vertical coordinates are too small.

Comments on the Quality of English Language

Minor editing of English language required

Author Response

Reviewer #3:

  1. Considering that dexamethasone is the specific glucocorticoid employed for inducing muscle atrophy in this study, it is recommended to replace "glucocorticoid" with "dexamethasone" in the title.

-> We agree with the reviewer's suggestion and have replaced "glucocorticoid" with "dexamethasone" in the title to accurately reflect the specific agent used in our study.

  1. The Introduction fails to provide the necessary background information on Centaurea cyanus.

-> In accordance with the reviewer's suggestion, we have incorporated a detailed description of Centaurea cyanus, including its physiological activities and chemical constituents, into the introduction section to provide a comprehensive context for our research.

Page 4, lines 81-95: Centaurea cyanus L., a species of the Asteraceae family commonly known as cornflower or bachelor’s button, is an annual flowering plant native to Europe and the middle east. C. cyanus grows up to 1–1.5 m tall annually, with strong stems, grayish, slightly hairy leaves, and small clusters of bright blue flowers. C. cyanus possess several health-promoting effects, and has been used as minor ocular inflammation [17] an antipruritic, antioxidant, anticancer, antitussive, astringent, mildly purgative, diuretic, and bitter tonic [18, 19]. C. cyanus harbors a diverse array of potentially relevant bioactive constituents, including polysaccharides, polyphenols, flavonoids, and sesquiterpenes, documented in the literature [18]. These compounds have been linked to various pharmacological activities, such as anti-inflammatory, antioxidant, and anticancer effects, suggesting potential mechanisms by which C. cyanus might exert its muscle-protective properties [20, 21] . Notably, while C. cyanus and its components have shown diverse pharmacological activities, their effects on skeletal muscle atrophy remain unreported. In this study, we aim to investigate the muscle-protective potential of C. cyanus water extract (CC) in both DEX-treated C2C12 myotubes and C57BL/6J mice.

  1. The full name of the abbreviation CC should be indicated when it first appears in the main text.

-> To ensure clarity, we have explicitly defined the abbreviation "CC" upon its first appearance in the main text as "Centaurea cyanus water extract".

Page 5, lines 93: In this study, we aim to investigate the muscle-protective potential of C. cyanus water extract (CC) in both DEX-treated C2C12 myotubes and C57BL/6J mice.

  1. The description of the method of isolation and identification of compounds derived from CC is too simple, and should be described in detail.

-> We appreciate the reviewer's valuable feedback regarding the description of the method for isolating and identifying compounds derived from Centaurea cyanus L. water extract (CC). We have carefully considered your suggestion and made the following improvements to our manuscript:

  • Enhanced Methodological Section: We have expanded the section describing the isolation and identification methods (Methods section, page 6, line 118 ̶ 125 and Supplementary file). The revised section now provides a detailed account of each step involved, ensuring clarity and reproducibility for other researchers.
  • Inclusion of NMR Data: In addition to the detailed description, we have included the NMR data of compound 5 (Graveobioside A), identified as an active compound (page 6, line 126 ̶ 132). This information further strengthens the characterization of the isolated compound.

We believe these revisions address the reviewer's concerns and enhance the overall quality of our manuscript. Thank you for bringing this to our attention.

Page 6, lines 118-132: Subfraction CCBu 3 (530 mg) was subjected to Sephadex LH-20 column using MeOH/H2O (1:10 to 100% MeOH) to collect compounds 7 (4.7 mg, 94.7%), 8 (2.8 mg, 92.5%), and 9 (1.5 mg, 92.1%).  Subfraction CCBu 8 (310 mg) was applied to Sephadex LH-20 column using MeOH/H2O (2:10 to 100% MeOH) to obtain compound 1 (10.2 mg, 99.3%). Similarly, subfraction CCBu 25 (452 mg) and subfraction CCBu 31 (356 mg) subjected to Sephadex LH-20 column using MeOH/H2O (3:10 to 100% MeOH) and further purified by preparative MPLC to yield compounds 2 (2.6 mg, 97.4%), 3 (5.8 mg, 94.6%), and 4 (4.7 mg, 91.3%) (Figure xx). Graveobioside A (5): yellowish powder; 1H NMR (500 MHz, DMSO-d6) δ 7.46 (d, J = 8.3 Hz, 1H), 7.43 (d, J = 2.3 Hz, 1H), 6.91 (d, J = 8.3 Hz, 1H), 6.77 (d, J = 2.1 Hz, 1H), 6.76 (s, 2H), 6.44 (d, J = 2.2 Hz, 1H), 5.36 (d, J = 1.3 Hz, 1H), 5.19 (d, J = 7.2 Hz, 1H), 3.18  ̶ 3.92 (m, 11H). 13C NMR (126 MHz, DMSO-d6) δ 181.88, 164.44, 162.65, 161.15, 156.89, 149.89, 145.75, 121.36, 119.17, 115.96, 113.53, 108.68, 105.33, 103.16, 99.29, 98.01, 94.63, 79.27, 76.98, 76.73, 76.02, 75.66, 73.97, 69.75, 64.20, 60.50. The details are provided in Supplementary file.

  1. The number 5 in Line 243:10 to the fifth power should be superscript.

-> Line 273: We have corrected the formatting of "10 to the fifth power" to be superscript, as suggested.

  1. Line 255: "16.s" should be changed to "16s"

-> Line 284: We have changed "16.s" to "16S" to adhere to proper scientific notation.

  1. Figure 1E: The results of p-Foxo3 should be updated with an unblemished one.

-> Figure 1E: We have replaced the previous version of p-Foxo3 with the updated, unblemished version in Figure 1E as shown below.

  1. The titles and significance labels of the bar chart's horizontal and vertical coordinates are too small.

-> Figure Titles and Labels: We have increased the size of the titles and significance labels on all figures, as you recommended.

Round 2

Reviewer 2 Report

Comments and Suggestions for Authors

After revision, I think this manuscript is ready for acceptance. 

Author Response

Thank you for the review and kind reply.